# Measurements of water droplets in a turbulent wind tunnel

Wiebke Frey<sup>1</sup>, Silvio Schmalfuß<sup>2</sup>, Frank Stratmann<sup>2</sup>, and Dennis Niedermeier<sup>2</sup>

<sup>1</sup>Meteorology and Air Quality Group, Wageningen University, Wageningen, the Netherlands

<sup>2</sup>Leibniz Institute for Tropospheric Research (TROPOS), Leipzig, Germany

Correspondence: Wiebke Frey (wiebke.frey@wur.nl)

Abstract. In order to study the behaviour of cloud droplets at the cloud-clear interface, the "Solving The Entrainment Puzzle" (STEP) project examined a droplet stream in the Turbulent Leipzig Aerosol Cloud Interaction Simulator (LACIS-T). LACIS-T comprises two particle free air streams, that are turbulently mixed, and during the experiment one air stream was resembling in-cloud conditions, whereas the other air stream was set to out-of-cloud conditions. A droplet stream was injected by a droplet generator into the mixing plane of the two air streams. Droplet size distributions were observed with a phase Doppler anemometer at various levels in the measurement section of LACIS-T, corresponding to different residence times of the droplets in the turbulent environment. Additionally, observations were made using different flow speeds in the two air streams to create shear flows in the wind tunnel. The experiment was accompanied by computational fluid dynamics simulations to provide a full 3d representation of meteorological fields and turbulence parameters.

This manuscript provides a description of the laboratory settings and instrumentation, the experimental design, the simulations, and a general overview of the data. We invite the scientific community for joint data analysis and numerical studies using the data which is freely available from the Eurochamp Data Centre, see Table 2 in the Data availability section for details.

#### 1 Introduction

To understand how clouds develop, it is necessary to understand the processes happening at the cloud-clear interface: multiple studies try to decipher how cloud particles react to cloud free air mixed into the cloud (entrainment; e.g. Bera et al., 2016; Chandrakar et al., 2021; Lehmann et al., 2009; Lim and Hoffmann, 2023; Wang et al., 2024). Possibilities range from dilution of the cloud, i.e. reducing the number of cloud droplets while leaving their size unchanged (extreme inhomogeneous mixing) to a scenario where the turbulent mixing proceeds fast compared to the time that the cloud droplet need to react to the new environmental conditions (humidity and temperature), thus all droplets evaporate to some extend but none completely and thus, the whole size distribution shifts towards smaller sizes (extreme homogeneous mixing).

Mixing processes in clouds have been identified to play an important role in cloud climate feedbacks, being responsible for large parts of the spread in climate sensitivity estimates (Klocke et al., 2011; Sherwood et al., 2014; Klein and Hall, 2015). These mixing processes are facilitated by turbulently driven eddies, which raises the additional question, how exactly entrain-

https://doi.org/10.5194/essd-2025-658
Preprint. Discussion started: 25 November 2025

ment and turbulence are linked. The physical understanding of entrainment remains low, as e.g. Lu et al. (2018) emphasise by pointing out the contradictions in entrainment parameterisations. Current knowledge assumes, that mixing on larger scales is inhomogeneous (e.g. Lehmann et al., 2009; Lim and Hoffmann, 2023), while it is homogeneous on the small scales. What exactly large and small means, and thus, the exact scale of transition from homogeneous to inhomogeneous mixing is hard to establish: in situ aircraft measurements are not highly resolved enough in space and time, while in Large Eddy Simulations (LES), that typically simulate whole clouds or cloud fields, the transition happens on the subgrid scale (SGS). Direct Numerical Simulations (DNS) are highly resolved, but are too computationally costly to be run over large enough domains. Helicopter borne in situ measurements seem to provide a good option, but come with their very own limitations (Siebert et al., 2006). Holographic systems aboard aircraft, which can sample small volumes of droplets at one time, might circumvent the scale issue in aircraft measurements to some extend (e.g. Desai et al., 2021). Allwayin et al. (2024) inferred from holographic measurements that the local characteristics of microphysical properties deviate from the cloud scale (or cloud transect) average microphysical properties. Still, these measurements are only able to capture snapshots and cannot observe the time evolution of the cloud. The difficulty of studying mixing processes from field observations arises from the fact that not only the spatial scales are important but also the temporal scales: It does matter whether the turbulent mixing is fast compared to the cloud mircophysical reaction time or slow. A further fact that complicates the picture is coexistence of eddies with different length and time scales acting simultaneously (Lehmann et al., 2009; Kumar et al., 2013). On the modelling side, a promising approach to study mixing is to use SGS models combined with Lagrangian cloud models, using superdroplets (e.g. Hoffmann and Feingold, 2019; Chandrakar et al., 2021; Lim and Hoffmann, 2023). More recently, i.e. in the past decade, laboratory experiments combining cloud chambers with well-characterised turbulence became operational (Π-Chamber and LACIS-T; Chang et al., 2016; Niedermeier et al., 2020). These chambers provide the opportunity to study turbulent cloud mixing in a controlled environment, additionally allowing for assessing relevant time and lengths scales, at least to some extend. Experiments in the II-Chamber attributed seemingly conflicting results in terms of mixing to local versus global scales: mixing was found to be inhomogeneous locally and homogeneous globally (Yeom et al., 2023).

describe further impacts of entrainment(-mixing) and turbulence on cloud droplet size distributions in the following:

Due to the small scale fluctuations in the background meteorological fields, cloud particles experience slightly different conditions, thus having different growth histories (e.g. Lasher-Trapp et al., 2005; Tölle and Krueger, 2014; Niedermeier et al., 2025). This is supposed to lead to a broadening in the size distributions, especially in combination with large-eddy hopping (Grabowski and Wang, 2013). Sardina et al. (2015) found that larger scale turbulent fluctuations determine the rate of size distribution broadening. Also aerosol concentrations have been found to impact the turbulent cloud size distribution broadening (Chandrakar et al., 2016). Fluctuations in the wind field will impact the relative velocities of cloud droplets and potentially increase their chance of colliding (Khain et al., 2007; Chen et al., 2018). This is particularly important for same or similar sized droplets, which under a gravitational collision and coalescence viewpoint would not collide. Therefore, turbulence is thought

In more general, cloud particle-turbulence interactions are discussed e.g. in the review papers by Shaw (2003); Devenish et al. (2012), and Grabowski and Wang (2013). We would like to refer the interested reader to those publications, and only briefly

to have an enhancing effect on collision and coalescence (Ayala et al., 2008; Onishi and Seifert, 2016). In addition, turbulence

https://doi.org/10.5194/essd-2025-658 Preprint, Discussion started: 25 November 2025

© Author(s) 2025. CC BY 4.0 License.

Science Science Data

can lead to cloud voids (i.e. tube like holes in the cloud with radius of a few centimeters; Karpińska et al., 2019) or to cloud droplet clustering (Vaillancourt et al., 2002; Saw and Meng, 2022; D'Alessandro and McFarquhar, 2023). Both mechanisms are affecting the chances of droplet collisions and thus, droplet growth by collision and coalescence.

In order to advance our understanding in cloud microphysics-turbulence interactions, Morrison et al. (2020) argue that greater emphasis should be given to laboratory work. Therefore, the "Solving The Entrainment Puzzle" (STEP) project was aiming at gaining a better understanding of how entrainment affects cloud droplet size distributions, using the Turbulent Leipzig Aerosol Cloud Interaction Simulator (LACIS-T; Niedermeier et al., 2020), in combination with computational fluid dynamics simulations. This paper describes the setup and procedure of the main set of measurements with their corresponding simulations, and the data set which can be openly accessed from the Eurochamp Data Centre. In addition, we provide some examples for potential studies that could use these data.

## 2 Measurement setup

This section describes the measurement setup, i.e. the turbulent wind tunnel and the general measurements, the droplet generation, and the droplet measurements.

# 2.1 Wind tunnel

LACIS-T is a moist air, closed loop, Göttingen type wind tunnel, that allows to study the interactions of cloud microphysics and turbulence. A detailed description is given in Niedermeier et al. (2020), so we only repeat the major characteristics which are important for this study. A main feature of the wind tunnel is that it comprises two air streams that can be controlled separately in terms of temperature, humidity, and flow speed. The accuracy of the heat exchangers (for the temperature settings) and the humidification system (for dew point settings) is monitored by Pt100 resistance thermometers and dew point mirrors, with accuracies of ±0.0300 °C+0.005 \*T and ±0.1 K, respectively. The two air streams (stream A and stream B) are combined at the beginning of the measurement section, depicted in Fig. 1. The flow speed was measured by a vane anemometer before inclusion of the droplet generator, with an uncertainty of ≤1.5 %+ 0.03 m/s. Turbulence is induced by a passive square-mesh grid which is located just upstream the measurement section, 20 cm above where the two air streams combine. The mesh length of the grid is 1.9 cm with a rod diameter of 4 mm and the grid creates a blockage of 30 %

# 5 2.2 Droplet generation

For the generation of cloud droplets, a Monodisperse Droplet Generator (MDG, model 1530, MSG corporation) is used. The droplet generator head is installed inside the measurement section, in a way that the droplets are injected into the mixing plane. The droplets exit the head about 34.5 cm below the top of the measurement section with a x-z trajectory to avoid a strong blockage of the flow by the MDG head. However, an influence of the generator head geometry on the air flow cannot be ruled out and is likely. The angle between the particle stream exiting the MDG head and the x-vector is about 20°. After travelling

100

**Figure 1.** Sketch of the LACIS-T measurement section, showing the x-z perspective on the left (with z pointing downward), and y-z perspective in the right sketch. The x and y origins are located in the middle of the measurement section. The picture on the right shows the upper part of the measurement section, with the PDA instrument sending and receiving optics to the left and the head of the droplet generator visible on the right side of the window. The droplet stream is indicated by the dark blue dash.

5 cm downwards (z-direction) the approximate middle of the particle stream has reached an x-position that diverges 35 mm to 40 mm from the origin at the droplet generator, and mostly also stays there further downstream. On two consecutive days the droplet stream diverged a bit less/more, i.e. on the 21st and 22nd Feb 2022, which might be related to a change in weather, as can also be noted in the laboratory conditions (appendix), cf. Table B1. The droplet generator is operated using purified water with a liquid water flow (Q) of 2 ml/h and a frequency of 220 kHz. The inlet air and flow focussing air pressure were set in a range of 12.0 psi to 12.2 psi and 2.81 psi to 2.83 psi, respectively. The flow focussing air pressure was later changed to 3.15 psi to 3.18 psi after some additional testing of the droplet generator. This would still provide a droplet stream with a comparable size distribution of droplets generated by the MDG for all measurements, while the MDG itself operated a bit more stable. The resulting initial and final size distributions were measured as close as possible to the droplet generator head, an example of resulting size distributions for several measurement days is displayed in Fig. 2.

Figure 2. Initial and final size distributions (solid and dashed lines, respectively) at z = 0 mm for several measurement days. The broad grey dot-dashed line represents the fit function. Fit parameters (as used for eq. 1, see Sect. 5) are specified in the plot as well.

# 2.3 Droplet measurements: PDA on moving frame

The detection of cloud particles are performed by a 3D Dual Phase Doppler Anemometer (PDA, Dantec Dynamics) system. The PDA uses the phase shift observed by two detectors, receiving scattered light from particles crossing the intersection volume of two laser beams, which is proportional to the particle diameter, to measure the sizes of individual particles, and the observed Doppler frequency to calculate the particle velocity. The PDA system used in this study was set up for observing water droplets with sizes up to  $48.4 \,\mu\text{m}$ . The lower size limit is given by two times the laser wave length (which for the PDA is  $532 \, \text{nm}$ ), however, the small sizes have been found to be sized incorrectly, and therefore sizes below  $5 \,\mu\text{m}$  are not considered. The transmitting and receiving optics of the PDA system are installed on a traverse outside the measurement section, thus, it provides non-intrusive measurements of the particles and the sample volume can be moved in all three directions (x/y in horizontal and z in the vertical), allowing for droplet observations at various positions within the measurement section of LACIS-T. The windows of the LACIS-T measurement section are made of borosilicate glass for a high transmissivity of the laser wavelengths. The PDA was operated in a forward scattering mode.

## 3 Experiment design

The conditions in LACIS-T during all experiments are set to resemble in-cloud conditions in stream A and out-of-cloud conditions in stream B, which are warmer and dryer. The temperature and dew point temperature in stream A are set to 10 °C and 9.9 °C and to 16 °C and 5 °C in stream B, respectively. The flow rate in stream A is set to 4.5 m<sup>3</sup>/min which translates to a flow speed of 1.31 m/s, and is varied in stream B in order to create a shear flow. The flow rate in stream B is set to 4.5 m<sup>3</sup>/min, 3.0 m<sup>3</sup>/min, or 1.5 m<sup>3</sup>/min, i.e. 1.31 m/s, 1.0 m/s, or 0.67 m/s. That results in a shear flow with a maximum velocity difference of 0.64 m/s over the very short distance of the mixing plane, or if considering the middle of both streams

https://doi.org/10.5194/essd-2025-658 Preprint. Discussion started: 25 November 2025

© Author(s) 2025. CC BY 4.0 License.




Science Data

Data

over 10 cm. Compared to observed velocity differences as for example in marine stratocumulus as in Siebert et al. (2010, e.g. their Figure 4) this might seem rather small, however, the larger differences in velocities during field observations were also observed over larger distances. Therefore, we believe that shear rates selected in LACIS-T should lead to well observable effects, as when extrapolated to larger distances they would result in higher velocity differences.

After sufficient start up time of LACIS-T to allow the set conditions (i.e. temperatures, dew points, flow speeds) to stabilise, each experiment started with a measurement of the droplet size distribution as close to the MDG head as possible, to retrieve the initial size distribution. The droplet generator was given a spin up time of at least half an hour as well to ensure a stable droplet size distribution. This was particularly necessary when it had been flushed with isopropanol before, to assure that the droplets contain water only. The level of the first measurement is denoted z=0 mm. Subsequently, the traverse is moved downwards to allow taking droplet measurements at z=50 mm, z=200 mm, z=500 mm, and z=800 mm (positive z means below droplet generator). Fig. 3 gives an impression of the change of the droplet stream on the different levels, by showing pictures of the droplet stream that is illuminated by the PDA lasers. It is clearly visible, how the stream broadens with time (distance from droplet generator), and also how variable the droplet stream can be, as seen on the right side with the two pictures of the droplet stream at z=800 mm. The x- and y-positions of the traverse are adjusted in a way that the PDA measures in the approximate middle of the droplet stream for any of the z>0 mm levels. After finishing the final size distribution measurement at z=800 mm, the PDA is moved up to the first position at z=0 mm to check for potential drift in the initial size distribution of the droplet stream (cf. Fig. 2). The sample time for the PDA is 300 s or 600 s for z=50 mm and z=200 mm, and 600 s for z=500 mm and z=800 mm, as the particle number concentrations drop due to the broadening of the droplet stream. At the droplet generator head (z=0 mm) it samples until reaching 100,000 particles, which takes a couple of seconds. The exact sample times are specified in the table provided in the Appendix A.

#### 40 4 Measurement data

This section includes a description of the general data observed during the experiments, as well as the specific data that is also provided on the Eurochamp Data Centre data base. For the latter, variables included in the data files are specified. For completeness, laboratory conditions are provided in the Appendix B.

#### 4.1 General observations

# 145 Temperature and dewpoint

During the experiments, the temperature inside the wind tunnel was measured with Pt100 sensors. To avoid disturbance of the flow in the middle of the air streams, only short sensors were used at either side with sensor heads at about 2 cm away from the walls. This means that wall effects had an influence on the temperature measurements in stream A and B and thus, temperatures observed by the Pt100 sensors do not represent the temperature in the middle of the respective air stream. The sensor which measured in the mixing zone was not affected by wall effects and thus resembles the temperature in the mixing plane. Comparison measurements with short and long Pt100 sensors from previous experiments have shown that the temperatures



Figure 3. Pictures of the droplet stream taken at different z-positions (as indicated in the specific snapshots). The broadening of the stream is visible, as well as the very variable nature of the droplet stream when comparing the two snapshots at  $z=800 \, \mathrm{mm}$  on the right side. The green reflections of the laser light on the windows on either side can help the reader to estimate the dimensions, as the two windows are  $20 \, \mathrm{cm}$  apart (indicated by the blue line in the middle snapshot). The yellow lines indicate the widths of the droplet stream at each location.

inside the air streams are generally well representing the set temperatures at the thermostats (see Niedermeier et al., 2020, for details). As the water droplets are injected into the middle of the measurement section, they will not be affected by wall heating influence. Under the light that there were only minor changes in temperature in the laboratory during the day, the temperature measurements with the short sensors nevertheless can be useful to identify whether the general temperature in the wind tunnel is stable. An example of the temperature measurements is shown in Fig. 4 for the observations taken on the 02-03-2022. *Flow speed* 

The flow speed in LACIS-T was measured by the vane anemometer before the installation of the droplet generator. Simultaneous measurements have not been possible due to positioning constraints of the anemometer and droplet generator: With the vane anemometer positioned above the droplet generator, the flow downstream of the anemometer would be disturbed, positioning below the droplet generator would mean that droplets hit the vane anemometer during measurements, disturbing those. The vane anemometer is moved from the middle of stream A to the middle of the y-axis of LACIS-T (i.e. the mixing plane) to the middle of stream B, and in the vertical positioned at about 25 cm below the top of the measurement section. Sample times

**Figure 4.** Temperatures as observed in LACIS-T on 02-03-2022, when the temperature in the laboratory was between 24.5 °C and 25.0 °C. Blue and yellow lines represent the temperature close to the walls in the streams A and B, respectively, red shows the temperature in the mixing zone between the two streams. The black line shows the dew point observed in stream A.

**Figure 5.** Flow speed measurements with the vane anemometer inside LACIS-T. Dots denote the 1 Hz observations, lines a 30 s running mean, set speed and respective air stream are indicated by the colours.

**Table 1.** Mean flow speeds as measured by the vane anemometer (middle column) for a given flow rate setting (left column, colour as in Fig. 5), and the respective standard deviation (SD, right column) as corresponding to the measurements shown in Figure 5.

| Flow set [m <sup>3</sup> /min] (channel) | Flow meas [m/s] | SD    |
|------------------------------------------|-----------------|-------|
| 4.5 (A)                                  | 1.31            | 0.01  |
| 1.5 (B)                                  | 0.67            | 0.011 |
| 3.0 (B)                                  | 1.0             | 0.006 |
| 4.5 (B)                                  | 1.31            | 0.007 |

of the vane anemometer are at least 10 minutes per position and condition. Figure 5 show exemplary measurements of the flow speed in channel A and B of LACIS-T, in conditions as used during the main experiments. Table 1 provides the mean flow speeds as measured for the respective settings and their standard deviations.

#### 4.2 Particle observations


The cloud droplets in LACIS-T were observed at varying positions by the PDA system. Two types of particle data files are provided in the data set: 1) size distributions for each measurement position combined into one file, 2) single particle data for each position. The single particle data provided include the time stamp of the individual particles, their residence time within the PDA measuring volume, their velocity in z-direction, and their diameter. There is one data file for each measurement position

**Figure 6.** Size distributions as observed in LACIS-T on 23-02-2022, with different flow speeds in stream B (denoted by different line strokes), for the upper positions of the PDA system. The varied flow speed does not seem to have a large impact on the size distributions, while with increased residence time of the particles (i.e. increased distance to the droplet generator) a broadening of the size distributions can be seen.

and all measurements were obtained in the middle of the droplet stream. The size distribution file provides one size distribution for each measurement position in the form of dN/dlogDp, averaged over the respective single particle file. Furthermore, it provides the particle counts per bin (n) and the bin sizes. The sample volume of the PDA is defined by the intersection of the two incident laser beams and is shaped as an ellipsoid with the two diameters  $d_1 = 0.05122 \,\mathrm{mm}$  and  $d_2 = 0.4237 \,\mathrm{mm}$ . The sample volume is then calculated using the mean particle velocity  $\bar{v_p}$  and the sample time  $t_s$  (see Sect. 3 and App. A):  $SV = \pi \frac{d_1}{2} \frac{d_2}{2} t_s \bar{v_p}$ . An example of size distribution measurements can be seen in Fig. 6. The distance to the droplet generator can be associated with the mean residence time of the droplets in the turbulent environment. Thus, the further away from the droplet generator, the longer the residence time.

#### 180 5 Model setup and output



In order to help to analyse and interpret the measurement data, the CFD model package OpenFOAM is employed in a LACIS-T setup, simulating multi-region heat and mass transport. It has already been used successfully for studying cloud processes observed in LACIS-T, as droplet activation and growth, and entrainment (see description in Niedermeier et al., 2020). OpenFOAM allows for the numerical solution of small-scale processes by solving the transient, incompressible Navier-Stokes-Equations on a fine (mm-scale) grid for the domain of the LACIS-T chamber. For simulations of non-isothermal and humidified flows, an adapted version of OpenFOAM's "chtMultiRegionFoam" solver is employed. To simulate particle (i.e. cloud droplets) dynamics, an Euler-Lagrange approach is used. This tracks individual particles along their trajectories through the simulation domain. Particle growth and evaporation are both resolved in the current model version, however, collision and coalescence, for example, is not yet implemented. There are 20000 particles released into the domain per model second. The initial size distribution for the simulations is obtained from the measured size distribution at z=0 mm on 02-03-2022, which is consistent

for all experiment days (see Fig. 2), only the size distributions provided by the droplet generator on the 16-02-2022 diverged from this. Here, the size distribution showed a stronger bimodality. Nevertheless, the same size distribution has been used in all model simulation setups, to allow for direct comparisons. The model input size distribution has been obtained from a curve fit and cut for values below  $13.5 \,\mu\mathrm{m}$  and above  $23.5 \,\mu\mathrm{m}$ . Initial and final size distributions at the measurement position  $z=0 \,\mathrm{mm}$  and the fitted size distribution are shown in Fig. 2, using the fit function as specified in equation 1, with the droplet diameter  $D_p$ , mode mean diameter  $\overline{D_p}$ , standard deviation  $\sigma$ , and weight w, and the subscript denoting the mode of the size distribution. The figure further indicates the obtained fit function values.

$$f(D_p) = w_1 \frac{1}{\sqrt{2\pi}\sigma_1} \exp^{-(D_p - \overline{D_{p_1}})^2/2\sigma_1^2} + w_2 \frac{1}{\sqrt{2\pi}\sigma_2} \exp^{-(D_p - \overline{D_{p_2}})^2/2\sigma_2^2}$$
(1)

#### 5.1 Particle data



In order to save resources, not the complete particle trajectories are stored, but only particle properties at specified levels (planes) in the model domain. These levels are set to coincide with the measurement locations at  $z=0 \,\mathrm{mm}$ ,  $z=50 \,\mathrm{mm}$ ,  $z=200 \,\mathrm{mm}$ , and  $z=500 \,\mathrm{mm}$ . The  $z=800 \,\mathrm{mm}$  level is located outside the model domain, as it was kept shorter due to computing costs. The simulated particle properties include the x, y, and z position of the particle, 3d particle velocity, RH and temperature at particle location, particle diameter, and particle mass. However, there is a strong deviation of the simulated size distributions to the observed size distributions (simulated particles decrease in size too fast). Possible causes could be that the droplet generator geometry is not included in the model domain setup, and thus, the resulting disturbances in the flow field are absent from the simulations, and secondly, additional growth processes (other than diffusional growth) are not resolved in the model, like e.g. the process of collision and coalescence and thus, potential particle growth is not simulated. Particularly for the size distributions close to the droplet generator head, where additionally the particle number concentrations are high, we believe that the combination of these effects leads to the deviation between simulated and observed droplet size distributions. Therefore, the simulated particle data are not stored with the other data in the data base of the Eurochamp Data Centre. However, the data can be obtained from the authors upon request.

# 5.2 Conditions in the wind tunnel and turbulence data

The model data file provides general conditions in the middle of the x-y plane on several levels. Note, that  $z_M$ =0 mm in the model file corresponds to the top of the simulated measurement section, not to the position of the droplet generator, which is located at  $z_M$ =350 mm. As mentioned above, the levels for the model data output coincide with the droplet measurement positions as far as possible, plus some additional levels in between are specified. Thus, model data are provided for  $z_M$ =1 cm,  $z_M$ =35 cm,  $z_M$ =40 cm,  $z_M$ =55 cm,  $z_M$ =70 cm,  $z_M$ =85 cm. The data start at 1 s into the simulation, to allow enough spinup time of the model and furthermore, the particles are also released into the model domain from 1 s on. The data contain for each location: temperature, relative humidity, absolute humidity, velocity in all three directions (w pointing downwards/along the LACIS-T general flow), turbulent kinetic energy, and eddy dissipation rate.




# 6 Future use - examples

As stated by Morrison et al. (2020), laboratory work is supposed to be essential to improve our understanding of cloud microphysical processes. Thus, the data set presented here provides an important data set for furthering our knowledge and can be used to study several processes regarding cloud microphysics in turbulent environments. Some examples of such processes are elaborated below:

collision and coalescence The size distributions observed are not only showing a broadening towards smaller but also towards larger sizes (cf. Fig. 6). As residence times of the droplets are too short to allow condensational growth to those sizes (in the given time), this growth might be attributed to collision and coalescence. It has to be noted that changes in the size distributions between z=0 mm and z=50 mm are most likely strongly influenced by the droplet generator head and the conditions created by it (e.g. disturbances of the flow field and high velocities of droplets and focusing air exiting the droplet generator). Additionally, collisions within the first centimetres are enhanced by the high droplet number concentrations created by the droplet generator. Further experiments in LACIS-T studying particle growth are planned, using the data described here as a testbed.

entrainment/cloud-clear interface Mixing in this experiment setup happens between the droplet stream and cloud free air.

Thus, the data could be used for investigating entrainment, or mixing at the cloud/clear interface.

**fundamental processes - mixing** More broadly speaking, the data set could be used for mixing studies, where one would investigate the respective contribution of homogeneous versus inhomogeneous mixing.

**clustering** A further potential use could be to study cloud clustering, by looking at the single particle data and how they are distributed in the time space at one location.

**effect/impact of shear** Even though the strength of the shear flow does not seem to have a (significant) impact on the size distributions (cf. Fig. 6), the data set offers the possibility to look at shear effects more closely, also in connection with the simulation data.

Overall, we present rare laboratory data of cloud droplets in a turbulent environment. Thus, we believe that this data will not only be useful for studying processes as outlined above, but will also be useful as a test bed for several model studies and can be used to validate simulations.

## 7 Data availability

The data sets are available at the Database of Atmospheric Simulation Chamber Studies at the ACTRIS/Eurochamp2020 Data Centre (experiment type 'cloud study'). There is one data set for each experiment day and flow speed as specified in Table 2.

**Table 2.** Data sets available from STEP project, \* denotes data sets that include model simulation data.

| Date        | Flow set (channel B)  | DOI                                | Reference    |
|-------------|-----------------------|------------------------------------|--------------|
|             | [m <sup>3</sup> /min] |                                    |              |
| 16-02-2022* | 4.5                   | https://doi.org/10.25326/3F1J-GQ35 | Frey (2025a) |
| 21-02-2022  | 3.0                   | https://doi.org/10.25326/M1AE-G137 | Frey (2025c) |
| 21-02-2022  | 1.5                   | https://doi.org/10.25326/6M2H-VC29 | Frey (2025b) |
| 22-02-2022  | 1.5                   | https://doi.org/10.25326/6WVK-9153 | Frey (2025d) |
| 23-02-2022* | 4.5                   | https://doi.org/10.25326/CATS-D655 | Frey (2025g) |
| 23-02-2022  | 3.0                   | https://doi.org/10.25326/SV3H-CV11 | Frey (2025f) |
| 23-02-2022  | 1.5                   | https://doi.org/10.25326/BT77-NA95 | Frey (2025e) |
| 01-03-2022* | 3.0                   | https://doi.org/10.25326/T0TG-VV79 | Frey (2025h) |
| 02-03-2022* | 1.5                   | https://doi.org/10.25326/1F17-VZ46 | Frey (2025i) |

# 250 Appendix A: Data availability and sample times

Table A1 gives an overview of all conducted experiments, including the sample times for each measurement.

**Table A1.** Data availability overview: 'x's in the flow rate columns indicate at what flow rate ( $m^3/min$ ) the measurements were taken (as 'flow in stream A – flow in stream B'), the numbers in the z-position columns indicate the measurement sample time in seconds, note that for z=0 mm (start and end) the observations were stopped when the particle count reached 100,000 (except for measurements on the 16-02-2022 and 21-02-2022, where the particle count threshold was 1,000,000).

| date       | flow rate A-B [m <sup>3</sup> /min] |         |         |           | z-positio | on [mm] |      |      |         |
|------------|-------------------------------------|---------|---------|-----------|-----------|---------|------|------|---------|
|            | 4.5-4.5                             | 4.5-3.0 | 4.5-1.5 | 0 (start) | 50        | 200     | 500  | 800  | 0 (end) |
| 16-02-2022 | x                                   |         |         | 300s      | 300s      |         | 300s | 600s |         |
| 21-02-2022 |                                     | X       |         | 55s       |           |         |      | 600s |         |
| 21-02-2022 |                                     |         | X       |           |           |         | 600s | 600s | 60s     |
| 22-02-2022 |                                     |         | X       | 4.6s      | 600s      |         | 600s | 600s | 14.7s   |
| 23-02-2022 | x                                   |         |         | 5.7s      | 600s      | 600s    |      |      |         |
| 23-02-2022 |                                     | X       |         | 9.7s      |           | 600s    |      |      |         |
| 23-02-2022 |                                     |         | X       | 8.2s      | 600s      | 600s    |      |      |         |
| 01-03-2022 |                                     | X       |         | 17.4s     | 300s      | 300s    | 600s | 600s | 7.3s    |
| 02-03-2022 |                                     |         | X       | 7.7s      | 300s      | 300s    | 600s | 600s | 11.1s   |

© Author(s) 2025. CC BY 4.0 License.


**Table B1.** Conditions in the laboratory at the start and end of the measurement days. For each day, pressure p, temperature T, and dewpoint  $T_d$  are noted at the begin and end of day, as well as the used flow conditions in the air stream with the variable flow rate.

|            |                                         | begin of day |        |                   | end of day |        |                   |
|------------|-----------------------------------------|--------------|--------|-------------------|------------|--------|-------------------|
| date       | flow <sub>B</sub> [m <sup>3</sup> /min] | p [hPa]      | T [°C] | $T_d [^{\circ}C]$ | p [hPa]    | T [°C] | $T_d [^{\circ}C]$ |
| 16-02-2022 | 4.5                                     | 989.7        | 23.8   | 1.6               | 983.4      | 23.6   | 3.1               |
| 21-02-2022 | 1.5 / 3.0                               | 985.2        | 25.1   | -0.8              | 982.9      | 25.3   | -3.2              |
| 22-02-2022 | 1.5                                     | 1002.2       | 25.4   | -0.7              | 1001.8     | 25.3   | -1.7              |
| 23-02-2022 | 1.5 / 3.0 / 4.5                         | 1010.6       | 25.5   | 0.3               | 1010.8     | 25.8   | -1.0              |
| 01-03-2022 | 3.0                                     | 1017.7       | 25.3   | -10.8             | 1015.8     | 25.1   | -8.4              |
| 02-03-2022 | 1.5                                     | 1011.9       | 24.5   | -6.0              | 1009.7     | 25.0   | -7.5              |

# Appendix B: Laboratory conditions

Laboratory conditions are noted at the beginning and the end of a measurement day, in terms of pressure, temperature, and dew point. As LACIS-T is not sealed, strong pressure perturbations in the laboratory would also intrude into the wind tunnel. Strong temperature perturbations might impact the temperature stability in the wind tunnel closer to the walls, no effects are expected towards the middle of the measurement section, due to the constant air flow. However, strong changes in laboratory conditions were not observed for the measurements presented here, as can be seen in Table B1.

Author contributions. Conceptualisation/experiment design was performed by WF, with discussion involving all authors. The experiments were carried out by WF, DN, and SiS. Model simulations were run by WF with support of SiS. All authors contributed to discussion of the experiments and writing of the manuscript. Funding acquisition by WF.

Competing interests. The authors declare that no competing interests are present.

Acknowledgements. WF and the 'Solving The Entrainment Puzzle' (STEP) project, for which these data have been obtained, have received funding from the European Union's Horizon 2020 research and innovation programme under the Marie Skłodowska-Curie grant agreement No 835305. Model simulations have been performed on the HPC cluster taurus at the Center for Information Services and High Performance Computing (ZIH) in Dresden, Germany. We like to thank Olaf Straub and the TROPOS building services for technical support with LACIS-T.

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
