# Peer review of "Measurements of water droplets in a turbulent wind tunnel"

_Earth System Science Data, 2025_

## Referee Comment (RC1)

**Review of "Measurements of water droplets in a turbulent wind tunnel."**

**General Comment:**

This manuscript aims to understand the behavior of cloud droplets in the entrainment-mixing process. It provides an adequate description of the experimental setup, numerical simulations, instrumentation, and the associated uncertainties, as well as the strengths and limitations of both the measured and simulated data. The dataset enables investigations of mixing processes under well-controlled boundary conditions, providing a valuable framework for enhancing the understanding of entrainment at the cloud–clear air interface.

Overall, the manuscript is well-written and contributes a valuable dataset to the atmospheric community. To enhance clarity, I have a few major and minor comments that can be addressed in a timely manner. I would recommend acceptance for the publication if the following comments are addressed properly.

**Major Comments**

**Comment 1:** Line 163: It mentions that measurement from the vane anemometer has also been performed 25 cm below the top of the measurement section. Can you include it in Fig. 5?

**Comment 2:** Line 189: How do 20000 particles released into the domain per model second compare with the droplet concentration injected from MDG in experiments?

**Comment 3:** Although the size distribution from the OpenFOAM simulation deviates significantly from the experiments. It would be helpful for data users if one snapshot of size distribution from the simulation at different heights could be included in the Main text or the Appendix, as you prefer.

**Minor comments:**

**Comment 1:** The title seems a little vague. It does not convey the message that which properties of water droplets have been measured. Include keywords such as microphysical properties or size distribution.

**Comment 2:** Line 27: Include the relevant citations for " it is homogeneous on small scales"

**Comment 3:** For clarity, mark the height of droplet injection in the left or middle subplot of Fig. 1.

**Comment 4:** Line 95-97: Also show the unit of pressure in hPa or a commonly used unit for the atmospheric science community.

**Comment 5:** Line 99: Does " as close as possible" represent z=0? Mention here also, although it is already in the Caption of Fig. 1 and later.

---

## Author Comment (AC1)

Reply to Review #1

We would like to thank the reviewer for the positive comments, which help to clarify certain points in the manuscript.
We provide a point by point answer to the major/minor comments below, comments by the reviewer in blue and our answers in black.
In addition, we realised that while in the figures it says z=XX cm, the text states z=XX mm. We have unified these occurrences to always use cm.

Major Comments
Comment 1: Line 163: It mentions that measurement from the vane anemometer has also been performed 25 cm below the top of the measurement section. Can you include it in Fig. 5?

We have included a statement about the position in the figure caption:
"The vane anemometer was positioned about 25cm below the top of the measurement section."

Comment 2: Line 189: How do 20000 particles released into the domain per model second compare with the droplet concentration injected from MDG in experiments?

As shown in Table A1, measurements at the outlet of the droplet generator were mostly performed until a count of 100,000 was reached. This was accomplished in about 5 seconds (for the "quickest" cases) or longer. Thus, 20,000 particles per second were chosen in the OpenFOAM setup. Comparing the count distribution for the 2022-03-02 experiment the initial size distributions compare relatively well, see Fig. R1. Some deviation is visible, caused by observed particles that lie outside the size range of the injected simulated particles.

[Figure]

Figure R1: Comparison of particle counts measured by the PDA (solid lines) and simulated by OpenFOAM (dashed lines) for the high shear flow experiment within one second. Colours indicate the position below the droplet generator.

Nevertheless, when comparing OpenFOAM and observed size distributions plotted as dN/dlogDp (Fig. R2), the peak in the initial OpenFOAM size distribution is about a factor of 5 lower than the peak at z=0cm of the observed size distribution. When injecting particles

in OpenFOAM, this is done at specified grid points and therefore, it is not possible, to exactly match the OpenFOAM and PDA sample areas, especially at z=0cm, which might cause the difference in dN/dlogDp seen here. Further downstream this is less of an issue as the droplet spray widens up and the sample area just covers a part of the area that the droplet stream would cover on that respective level. Deviations of observed and simulated size distributions are mainly caused by the different size range covered.

[Figure]

Figure R2: Comparison of size distributions measured by the PDA (solid lines) and simulated by OpenFOAM (dashed lines) for two experiments with different flow settings, the low shear flow experiment on the left and high shear flow experiment on the right. Colours indicate the position below the droplet generator.

We added in the manuscript
"There are 20000 particles released into the domain per model second, which roughly corresponds to the 100,000 particles observed during the shortest initial measurements at z=0cm that took about 5 seconds (see Table A1)."

Comment 3: Although the size distribution from the OpenFOAM simulation deviates significantly from the experiments. It would be helpful for data users if one snapshot of size distribution from the simulation at different heights could be included in the Main text or the Appendix, as you prefer.

Thank you for this comment, we included a size distribution plot including all simulations in the appendix and added a reference to the appendix in the main text.

"The simulated size distributions are shown for completeness in Appendix C"

"Appendix C: Simulated size distributions
Simulated size distributions for the three different flow settings as used during the experiments on 16-02-2022, 01-03-2022, and 02-03-2022 are shown in Figure C1. The figure also includes the size distribution for one additional simulation for the same flow experiment (16-02-2022), where a slight deviation of 0.05 m/s was imposed on the flow speed in channel B, to identify whether a potential deviation in flow speed during the measurements would have a noticeable effect on the turbulence characteristics and potentially on the droplet size distributions.

[Figure]

Figure C1. Simulated size distributions for all flow setting used during the experiments (and one sensitivity on small flow deviation), sampled at the same locations downstream the droplet generator as in the experiments. The size distribution at z=800 mm is located outside the model domain, and thus, not simulated."

Minor comments:
Comment 1: The title seems a little vague. It does not convey the message that which properties of water droplets have been measured. Include keywords such as microphysical properties or size distribution.

Thanks for the suggestion. We changed the title to:
"Measurements of water droplet size distributions in a turbulent wind tunnel"

Comment 2: Line 27: Include the relevant citations for " it is homogeneous on small scales"

It's actually the same as for the inhomogeneous mixing, thus we moved the citations to the end of the sentence.

Comment 3: For clarity, mark the height of droplet injection in the left or middle subplot of Fig. 1.

We included an indication of the droplet generator height:

[Figure]

Fig. 2

Comment 4: Line 95-97: Also show the unit of pressure in hPa or a commonly used unit for the atmospheric science community.

we added the pressures in mbar:
"The inlet air and flow focussing air pressure were set in a range of 12.0 psi to 12.2 psi and 2.81 psi to 2.83 psi, respectively (i.e. 827.4 mbar to 841.2 mbar and 193.7 mbar to 195.1 mbar). The flow focussing air pressure was later changed to 3.15 psi to 3.18 psi (217.2 mbar to 219.3 mbar) after some additional testing of the droplet generator."

Comment 5: Line 99: Does " as close as possible" represent z=0? Mention here also, although it is already in the Caption of Fig. 1 and later.

added
"as close as possible to the droplet generator head (z=0 mm)"

---

## Author Comment (AC2)

Reply to Review #2

We would like to thank the reviewer for the positive comments, which help to clarify certain points in the manuscript.
We provide a point by point answer to the comments below, comments by the reviewer in blue and our answers in black.
In addition, we realised that while in the figures it says z=XX cm, the text states z=XX mm. We have unified these occurrences to always use cm.

- The data set could also be useful for model inter-comparison projects (e.g., a case for the next International Cloud Modeling Workshop) and evaluating different subgrid turbulence-microphysics interaction schemes. The authors should consider highlighting that in the manuscript.

Thanks for pointing this out! We have expanded the "Future use" section as follows:

Thus, we believe that this data will not only be useful for studying processes as outlined above, but will also be useful as a test bed for several model studies, both on individual level and for model inter-comparison projects.  Additionally, it can be used to validate simulations.

- L35: The authors could also mention the CloudKite measurements as another unique way to collect high-resolution data for entrainment-mixing studies.

Thanks for the suggestion, we have included a sentence on CloudKite as well.

"Similarly, tethered balloon systems using helikites like the CloudKite (Schlenczek et al., 2026, Stevens et al., 2021) can provide a platform for highly resolved observations. However, as ground-anchored systems, they cannot actively target regions of interest horizontally and are limited by payload and maximum ceiling."

- L82: It's confusing to see uncertainties reported as % +- some value. Do you mean '1.5% | v| + 0.03 m/s' here?

Yes, the measurement uncertainty equals  ± (≤ 1.5 % of the measured value + 0.03 m/s), thus, e.g. for a flow speed measurement of 1m/s it is ± 0.045m/s.
We adapted:
"... with an uncertainty of  ± (≤ 1.5 % (measured value) + 0.03 m/s)"

- L179: Can the authors add a paragraph describing turbulence statistics in the measurement section? It would also be helpful to show PDFs of temperature, water vapor, and velocity fluctuations (and their moments) at different vertical locations in the mixing zone.

Direct measurements of fluctuations are unfortunately not available (partly due to lack of needed instrumentation, e.g. for water vapour fluctuations or incompatibility to use instrumentation for fluctuations together with the droplet stream).

The general turbulence characteristics of LACIS-T have been described in detail in Niedermeier et al. (2020), who also show the ability of the OpenFOAM setup to reliably simulate the flow characteristics. Therefore, we expect the model results to properly represent the fluctuations also in our experiments. We added to the manuscript in Sect. 4.1:

*"Turbulence characteristics*

Measurements of fluctuations to determine the turbulence statistics of the flow are unfortunately unavailable for this set of measurements. However, turbulence characteristics of LACIS-T based on observations (and simulations) are described in Niedermeier et al. (2020), who have also shown that the OpenFOAM setup is able to reproduce the characteristics in its simulations. Thus, we provide some information on the turbulence characteristics in Sect. 5.2."

In Sect. 5.2 we added:

"Figure 7 shows some of the simulated turbulence statistics (eddy dissipation rate ($\epsilon$), turbulent kinetic energy (TKE), Taylor microscale, and Taylor Reynolds number) in the mixing plane of LACIS-T."

[Figure]

Figure 7. Turbulence statistics versus position below the droplet generator as simulated with OpenFOAM, from left to right: eddy dissipation rate ($\epsilon$), turbulent kinetic energy (TKE), Taylor microscale, and Taylor Reynolds number. Different colours and symbols denote the different flow settings.

As can be seen from Fig. R1, with increasing shear, the velocity fluctuations become stronger (increasing widths of distribution), and thus, TKE (c.f. Fig. 7). Also the mixing between the two air streams become stronger with increasing shear, one could sense a small decrease in width of the RH and temperature distributions, which however, are also impacted by a less symmetric flow with increasing shear. The somewhat strange behaviour of TKE, Taylor microscale, and Talyor Reynolds number for the 0.67m/s at 20cm and 1.0m/s at 35cm points might be caused by formation of a standing vortex or by effects from the slit between the two air streams at the point where they combine (which contains the aerosol inlet).

We furthermore added to Sect. 5:
" Model simulations are run for all three flow settings (no shear/low shear/high shear) resembling the measurements on the 16-02-2022, 01-03-2022, and 02-03-2022. One additional simulation for the no shear case was run, where a slight deviation of 0.05m/s was imposed on the flow speed in channel B, to identify whether a potential deviation in flow speed during the measurements would have a noticeable effect on the turbulence characteristics and potentially on the droplet size distributions."

[Figure]

Fig. R1: Density distributions of simulated temperature (upper left), relative humidity (upper right), and velocity fluctuations in z-direction (bottom), for all simulated flow speed settings (see different line styles) and at all simulated positions below the droplet generator (z-position indicated in the panels).

- L204: The sentence starting with 'However' seems abrupt. I can't see any statement above that you are contrasting. Is there something missing?

Thanks for the observation, we changed the "However" to
"It has to be noted that"

- Please discuss the assumptions underlying the simulation setup in more detail. For example, does the condensation growth model include the curvature and solute effects (or a pure water drop assumed)?

The general model setup is described in detail in Niedermeier et al. (2020), we make the connection more explicit in the revised manuscript. A brief addition here to the original manuscript:

For the simulation of the non-isothermal flow the conservation equations for mass, momentum, energy, and water vapour content are solved (using the adapted version of OpenFOAM's "chtMultiRegionFoam" solver). For simulating turbulence we employ the dynamic k-equation Large Eddy Simulation model.
For the particle phase: The droplet activation is calculated according to Köhler theory (i.e. including curvature and solute effects, however in our experiments we use and simulate pure water droplets). Given the size of the droplets in our experiments, the curvature effect on the particle growth becomes negligible.

We changed the manuscript accordingly:
"For simulations of non-isothermal and humidified flows, an adapted version of OpenFOAM's "chtMultiRegionFoam" solver is employed, which solves the conservation equations for mass, momentum, energy, and water vapour content. Turbulence is simulated using the dynamic k-equation Large Eddy Simulation (LES) model. For further details see Niedermeier et al. (2020). To simulate particle (i.e. cloud droplets) dynamics, an Euler-Lagrange approach is used. This tracks individual particles along their trajectories through the simulation domain. Droplet growth (including Köhler theory) and evaporation are both resolved in the current model version, with a two way transport of water vapour between the fluid and particle phase. Additional growth by collision and coalescence and ice microphysics are not (yet) implemented."

Additionally, we detected one error in the introduction of the OpenFOAM model, in our case, we solve the compressible (not the incompressible) Navier-Stokes-Equations. We changed the text accordingly.

- L227-229: Growth by drop collision-coalescence - But the droplet concentration distribution (left plot in Fig. 6) shows hardly any change for the large droplet tail concentrations. Normalized distributions (PDFs) can be misleading when comparing tails if the mode decreases more than the tails.

True, this is why we chose to include both – the dN/dlogDp and the normalised (density) plot here. As stated in the following of the lines you mention, the most part of collision-coalescence is possibly happening in the first few centimetres downstream the droplet generator (thus between z=0mm and z=50mm), and here the shift in tail is clearly visible. We agree, that further downstream the effect is hardly visible, also given the lower droplet number concentrations, only few counts would be expected. Thus, a more statistical analysis would be needed, which is hard to convey in a size distribution plot. However, from 0cm to 5cm (blue to red curve) the broadening towards larger sizes is clearly visible.

- L234: As I mentioned earlier, it would be very useful to discuss the turbulence statistics relevant for collision-coalescence (e.g., turbulent dissipation rates, Taylor's micro-scale Reynolds number, Stokes number, PDFs of velocity fluctuations, etc.).

See answer to the turbulence characteristics above. In addition, we added a sentence in the section on particle observations (Sect. 4.2):

"The mean Stokes numbers for the droplets are for all settings < 0.053, calculated following Devenish et al. (2012), using the measurements to retrieve the droplet characteristic inertial response time, and the simulations for retrieving the Kolmogorov timescale."

References:

Devenish, B. J., Bartello, P., Brenguier, J.-L., Collins, L. R., Grabowski, W. W., IJzermans, R. H. A., Malinowski, S. P., Reeks, M. W., Vassilicos, J. C., Wang, L.-P., and Warhaft, Z.: Droplet growth in warm turbulent clouds, Quarterly Journal of the Royal Meteorological Society, 138, 1401–1429, https://doi.org/10.1002/qj.1897, 2012.

Niedermeier, D., Voigtländer, J., Schmalfuß, S., Busch, D., Schumacher, J., Shaw, R. A., and Stratmann, F.: Characterization and first results from LACIS-T: a moist-air wind tunnel to study aerosol–cloud–turbulence interactions, Atmospheric Measurement Techniques, 13, 2015-2033, https://doi.org/10.5194/amt-13-2015-2020, 2020

Schlenczek, O., Nordsiek, F., Brunner, C. E., Chávez-Medina, V., Thiede, B., Bodenschatz, E., and Bagheri, G.: Airborne measurements of turbulence and cloud microphysics during PaCE 2022 using the Advanced Max Planck CloudKite Instrument (MPCK+), Earth System Science Data Discussions, 2025, 1–29, https://doi.org/10.5194/essd-2025-112, 2025.

Stevens, B. et al.: EUREC[4]A, Earth System Science Data, 13, 4067–4119, https://doi.org/10.5194/essd-13-4067-2021, 2021.